# Pyranoanthocyanins Interfering with the Quorum Sensing of *Pseudomonas aeruginosa* and *Staphylococcus aureus*

**DOI:** 10.3390/ijms22168559

**Published:** 2021-08-09

**Authors:** Patrícia Coelho, Joana Oliveira, Iva Fernandes, Paula Araújo, Ana Rita Pereira, Paula Gameiro, Lucinda J. Bessa

**Affiliations:** LAQV-REQUIMTE, Department of Chemistry and Biochemistry, Faculty of Sciences, University of Porto, 4169-007 Porto, Portugal; patriciaines20@gmail.com (P.C.); jsoliveira@fc.up.pt (J.O.); iva.fernandes@fc.up.pt (I.F.); prauinha_araujo@hotmail.com (P.A.); armartinspereira@hotmail.com (A.R.P.); agsantos@fc.up.pt (P.G.)

**Keywords:** pyranoanthocyanins, biofilms, *Pseudomonas aeruginosa*, quorum sensing, *Staphylococcus aureus*, reverse transcription-quantitative polymerase chain reaction (RT-qPCR)

## Abstract

Bacterial quorum sensing (QS) is a cell-cell communication system that regulates several bacterial mechanisms, including the production of virulence factors and biofilm formation. Thus, targeting microbial QS is seen as a plausible alternative strategy to antibiotics, with potentiality to combat multidrug-resistant pathogens. Many phytochemicals with QS interference activity are currently being explored. Herein, an extract and a compound of bioinspired origin were tested for their ability to inhibit biofilm formation and interfere with the expression of QS-related genes in *Pseudomonas aeruginosa* and *Staphylococcus aureus*. The extract, a carboxypyranoanthocyanins red wine extract (carboxypyrano-ant extract), and the pure compound, carboxypyranocyanidin-3-*O*-glucoside (carboxypyCy-3-glc), did not cause a visible effect on the biofilm formation of the *P. aeruginosa* biofilms; however, both significantly affected the formation of biofilms by the *S. aureus* strains, as attested by the crystal violet assay and fluorescence microscopy. Both the extract and the pure compound significantly interfered with the expression of several QS-related genes in the *P. aeruginosa* and *S. aureus* biofilms, as per reverse transcription-quantitative polymerase chain reaction (RT-qPCR) results. Indeed, it was possible to conclude that these molecules interfere with QS at distinct stages and in a strain-specific manner. An extract with anti-QS properties could be advantageous because it is easily obtained and could have broad, antimicrobial therapeutic applications if included in topical formulations.

## 1. Introduction

Antimicrobial resistance (AMR) is one of the most concerning global public health threats of today, resulting in a significant increase in mortality and morbidity due to bacterial infections that were once treatable but now are life threatening, entailing huge economic costs [1,2].

Over the last years, researchers have been striving to find and develop new antibiotics to effectively treat multidrug-resistant bacterial infections, particularly those caused by the so-called priority pathogens, which are bacterial species with critical, high, and medium antibiotic resistance [3]. The species considered critical include *Acinetobacter baumannii, Pseudomonas aeruginosa* and various *Enterobacteriaceae* (including *Klebsiella pneumonia*, *Escherichia coli*, *Serratia* spp., and *Proteus* spp.). However, a key challenge has not yet been timely addressed, and that is that sooner or later pathogens develop or acquire resistance to the new drugs by selective pressures [4,5].

Hence, it is urgent to develop alternative approaches to tackle multidrug-resistant bacterial infections. A recently explored strategy regards the use of anti-virulence agents, also known as drugs that can block the virulence factors of the pathogen, impairing its ability to cause infection [4,6]. These drugs can, in theory, control bacterial infections not by killing the pathogen, but by affecting other pathways and mechanisms, such as their communication system, known as quorum sensing (QS). Quorum sensing is a cell-cell communication system that coordinates several bacterial mechanisms and behaviors in a density-dependent manner, allowing bacteria to share information and modulate gene expression accordingly [7,8,9]. In fact, QS was shown to be involved in both the formation of biofilms and the expression of virulence factors, two key mechanisms that enable bacteria to colonize and harm the host [10,11]. Thus, inhibiting QS could reduce virulence factor production and biofilm formation, increasing consequently the sensibility of the pathogen to antimicrobials and allowing the host immune system to tackle the infection more easily [6,12,13]. Given the importance of QS in the pathogenesis, QS inhibitors (QSI) can represent a promising approach as novel, anti-virulence drugs. Moreover, compared with conventional antibiotics, anti-virulence agents and QSI are less likely to cause the selective pressures that induce the emergence of resistance mechanisms [4].

Quorum sensing inhibitors can be natural or synthetic compounds, enzymes or antibodies [14,15,16]. Among the natural compounds, some polyphenols, such as anthocyanins, have been studied for their activity as QSI [14,17]. Anthocyanins are secondary metabolites produced by many plants and have been proven to inhibit QS-dependent phenotypes, such as violacein production by *Chromobacterium violaceum* [18,19], biofilm formation and virulence factor production by *P. aeruginosa*, *Shewanella baltica* and *K. pneumoniae* [18,20,21,22,23]. Recently, we have also shown the ability of anthocyanin-related structures, including pyranoanthocyanin extracts and their isolated compounds, to inhibit the biofilm formation of *P. aeruginosa* and *Staphylococcus aureus* [24].

Gram-negative and Gram-positive bacteria communicate using different QS systems, but the basic mechanisms are identical. The communication is achieved by the production, detection and response to small extracellular signaling molecules, known as autoinducers (AIs), that are sensed by surrounding bacteria [25,26]. The main difference lies on the AIs produced. Gram-negative bacteria use acyl homoserine lactones (AHLs) as signal molecules, while Gram-positive microorganisms communicate using autoinducing peptides (AIPs) [25,26]. Bacteria monitor the AI concentration to reach high cell density, resulting in a synchronized response by the bacterial population [27].

*P. aeruginosa* highly depends on QS to induce infection, having four QS systems as follows: LasI/LasR, RhlI/RhlR, Pseudomonas quinolone signal (Pqs) and Integrated QS signal (Iqs), with each system producing and responding to a specific signal molecule, 3-oxo-C12-homoserine lactone (3OC12-HLS), butanoyl homoserine lactone (C4HLS), 2-heptyl-3-hydroxi-4-quinolone (PQS) and 2-(2-hydrodyphenyl)-thiazole-4-carbaldehyde (IQS), respectively [26,27,28,29]. These systems are hierarchically organized, with the LasI/LasR system at the top, activating all the other systems. On the other hand, the RhlI/RhlR system represses the Pqs system, while the Pqs activates RhlI/RhlR [30,31]. The interplay between the four systems results in the production of a variety of virulence factors [27,28,32,33].

*The S. aureus* pathogenesis depends on the production of virulence factors that are partially controlled by QS, via a two-component system encoded by the accessory gene regulator (agr) locus [10,34]. The AIP is synthesized from *agrD* and transported to the extracellular medium [10]. Upon AIP detection, the histidine kinase receptor activates a signaling cascade that leads to AgrA phosphorylation and the subsequent activation of two divergent promoters, P2 and P3, that result in the expression of a regulatory RNA, RNAIII, which in turn induces the expression of several virulence factors [10,27,34].

The aim of this study was to explore a carboxypyranoanthocyanins extract obtained from the reaction of red wine anthocyanins with pyruvic acid, and a pure carboxypyranocyanidin-3-*O*-glucoside, which have previously shown to effectively hamper the biofilm formation of *P. aeruginosa* ATCC 27853 and *S. aureus* ATCC 29213 [24], for their: (i) effect in the biofilm formation of the *P. aeruginosa* and *S. aureus* multidrug-resistant (MDR) isolates; (ii) ability to interfere with the expression of the QS-related genes of the *P. aeruginosa* and *S. aureus* reference strains and MDR isolates by reverse transcription-quantitative polymerase chain reaction (RT-qPCR); and (iii) toxicity in the *Galleria mellonella* larvae model.

## 2. Results

### 2.1. Antimicrobial and Biofilm Inhibition Activities of Carboxypyrano-ant Extract and CarboxypyCy-3-glc against P. aeruginosa and S. aureus MDR Isolates

The carboxypyrano-ant extract and carboxypyCy-3-glc had previously not shown antibacterial activity against several Gram-positive and Gram-negative reference strains [24]. Now, in this study, the MIC and MBC values were obtained against the MDR isolates of *P. aeruginosa* and *S. aureus* (Pa3, PA004, SA007 and SA011), and are presented in Table 1.

Equally, neither the carboxypyrano-ant extract nor the carboxypyCy-3-glc presented antibacterial activity against any of the isolates tested (MIC > 512 μg/mL). Subsequently, the ability of the carboxypyrano-ant extract and carboxypyCy-3-glc in interfering with the biofilm formation by those MDR isolates was also evaluated by using the crystal violet method. The results regarding the *P. aeruginosa* and *S. aureus* isolates are displayed in Figure 1.

The extract and the compound did not affect the biofilm formation by either of the two MDR *P. aeruginosa* isolates, at any of the tested concentrations (Figure 1A).

In the MDR *S. aureus* isolates, the carboxypyrano-ant extract significantly decreased the biofilm formation at 256 and 64 μg/mL, and in the presence of carboxypyCy-3-glc, significantly less biofilm biomass was quantified at all concentrations (Figure 1B).

### 2.2. Impact of Carboxypyrano-ant Extract and CarboxypyCy-3-glc on the Biofilm Formation—Qualitative Microscopic Analysis

To further explore the phenotypic effects of the carboxypyrano-ant extract and carboxypyCy-3-glc on biofilm formation, the Live/Dead-staining technique coupled with fluorescence microscopy was performed on the biofilms of the MDR *P. aeruginosa* (PA004 and Pa3) and *S. aureus* isolates (SA007 and SA011). Regarding the biofilms of *P. aeruginosa*, we could observe a strong biofilm formation by both strains, either in the absence (controls) or presence of the extract and pure compound at 64 μg/mL (Figure 2).

Strong biofilms were formed by these two strains, and no significant differences could be pinpointed between the biofilms formed in the presence and in the absence of the compounds, confirming the results that were obtained in the previous assay.

Concerning the *S. aureus* biofilms, images obtained undoubtedly showed the effect of these compounds in hampering the biofilm formation (Figure 3). In the presence of either the carboxypyrano-ant extract or carboxypyCy-3-glc nearly any biofilm was formed by the *S. aureus* isolates, with only a few cells and aggregates adhered to the surface.

### 2.3. Interference of Carboxypyrano-ant Extract and CarboxypyCy-3-glc with the QS-Mediated Production of Violacein by C. violaceum

Quorum sensing inhibition activity of the carboxypyrano-ant extract and carboxypyCy-3-glc was primarily screened using *C. violaceum* ATCC 12472, which produces a purple pigment, violacein, under the regulation of a QS system. Thus, the inhibition of violacein production could indicate a potential interference with the QS communication system.

The quantitative determination of violacein production by HPLC (Figure 4) showed that carboxypyCy-3-glc appears to influence violacein production in a dose dependent manner, while the carboxypyrano-ant extract (at both 64 and 256 μg/mL) not only did not inhibit the production of violacein, but it also stimulated its production.

### 2.4. Interference of Carboxypyrano-ant Extract and CarboxypyCy-3-glc with the Expression of QS-Regulated Genes in P. aeruginosa and S. aureus Biofilms

As both individual carboxypyrano-ant extract and pure carboxypyCy-3-glc showed promising results in phenotypically inhibiting biofilm formation by *S. aureus* isolates, relative gene expression of several QS-related genes was quantified by RT-qPCR for *S. aureus* ATCC 29213, SA007 and SA011 biofilms formed in the presence of the carboxypyrano-ant extract or carboxypyCy-3-glc, at 64 μg/mL. Moreover, even though no significant effects were observed in the biofilm formation by the *P. aeruginosa* strains, the individual effect of both the extract and compound in the expression of genes associated with *P. aeruginosa* ATCC 27853 and Pa3 QS was also evaluated.

As presented in Figure 5A, the carboxypyrano-ant extract significantly downregulated the transcription of *lasI* and *rhlI* and upregulated the component of the Pqs system, *pqsE*, in *P. aeruginosa* ATCC 27853, whereas, in the Pa3 isolate, the extract caused upregulation of *lasR* expression. CarboxypyCy-3-glc also downregulated *lasI* and *rhlI* expression and in addition upregulated *pqsA* and *pqsE* expression in *P. aeruginosa* ATCC 27853; while in Pa3, it downregulated *rhlI* and *rhlR* expression (Figure 5B).

A schematic representation of the effects of the carboxypyrano-ant extract and carboxypyCy-3-glc on the QS systems of *P. aeruginosa* ATCC 27853 and/or Pa3 is presented in Figure 6.

The effects of the carboxypyrano-ant extract in the expression of *S. aureus* QS-related genes are shown in Figure 7A. 

In *S. aureus* ATCC 29213, the carboxypyrano-ant extract significantly decreased the transcription of *agrA*, and that, as expected, led to downregulation of downstream genes, such as RNAIII and *hla*; however, not in a statistically significant way. In the SA007 isolate, treatment with the carboxypyrano-ant extract led to upregulation of *sarA* expression and concomitant downregulation of *hla* and *ica*, while in SA011, the extract decreased the expression of *agrA*, *sarA* and *hla*. As shown in Figure 7B, carboxypyCy-3-glc activated *sarA* and RNAIII transcription and downregulated *hla* and *ica* expression in *S. aureus* ATCC 29213, upregulated the expression of *sarA*, RNAIII and *hla* in SA007, and decreased *agrA* and *hla* transcription in SA011.

A schematic representation of the effects of the carboxypyrano-ant extract and carboxypyCy-3-glc on the QS system of *S. aureus* ATCC 29213 and/or SA007 and SA011 is shown in Figure 8.

### 2.5. Toxicity and Assessment of Carboxypyrano-ant Extract and CarboxypyCy-3-glc in G. mellonella

Due to its numerous advantages as an in vivo model, *G. mellonella* larvae were used to assess the toxicity of the carboxypyrano-ant extract and carboxypyCy-3-glc. The toxicity of the extract and pure compound was assessed at two concentrations, 25 and 50 mg/kg, which were considerably higher than the concentrations used in the biofilm formation assay and in the gene expression assays. As shown in Figure 9, carboxypyCy-3-glc showed a substantial toxic effect at 50 mg/kg on the larvae, whilst none of the remaining conditions tested (carboxypyCy-3-glc at 25 mg/kg and carboxypyrano-ant extract at 50 and 25 mg/kg) appeared to have significant toxicity in the *G. mellonella* model.

## 3. Discussion

The carboxypyrano-ant extract and carboxypyCy-3-glc did not affect the growth of any of the multidrug-resistant isolates assayed. Nevertheless, they both phenotypically hampered the biofilm formation by MRSA isolates, while not affecting the biofilm formation by MDR *P. aeruginosa* isolates. Given the fact that the biofilm formation of *P. aeruginosa* ATCC 27853 has been affected by the carboxypyrano-ant extract and carboxypyCy-3-glc [24], these results may suggest that either the compounds’ target is altered, inaccessible or absent, or regulated differently in MDR *P. aeruginosa* biofilms. Hypothesizing that the target of the compounds is the QS system, as we anticipate, and knowing that the QS mechanism involves the production, release, and detection of chemical signaling molecules, it is likely that differences between strains may occur in this system regulation upon the presence of these compounds [35,36,37].

The total content of carboxypyranoanthocyanins in the carboxypyrano-ant extract is nearly 20%, although there is a great contribution of other constituents (approximately 10% proteins, 5% lipids (palmitic, oleic and stearic acids) and approximately 2% simple sugars). It is worth to mention that besides the reduced phenolic content, the huge amount of polymeric anthocyaninic structures (not detected in any of the chromatographic or colorimetric assays) likely present in this extract may also account for the observed inhibitory effect on biofilm formation, acting synergistically. Besides, it was described that some compounds present in red wine, such as other flavonoids like flavonols derivatives (myricetin-*O*-(*O*-galloyl)arabinoside, myricetin-3-*O*-arabinoside, quercetin 3-methoxyhexoside and quercetin 3-*O*-glucuronide tentatively identified) present in our carboxypyrano-ant extract, also have anti-virulence activity and affect biofilm formation, and, therefore, they could also contribute to the inhibition of biofilm formation observed in the presence of the carboxypyrano-ant extract [19,38]. Moreover, the role of the other components of the extract cannot be dismissed as well, since fatty acids may act as inhibitors of biofilm development and virulence at low concentrations [39], and yeast mannoproteins also present in the extract may equally contribute to the inhibitory effect on biofilm formation [40]. Therefore, the observed effect may result from the joint contribution of the different components.

Next, we have used a natural indicator strain, *C. violaceum*, for QSI screening, which involves the quantification of violacein produced by the strain. CarboxypyCy-3-glc inhibited violacein production in a dose dependent manner, which corroborates with the results of biofilm formation obtained by the crystal violet assay in *P. aeruginosa* ATCC 27853 [24]. However, interestingly, herein the carboxypyrano-ant extract stimulated violacein production (at both 64 and 256 μg/mL). This observation may be explained by the fact that although they are both Gram-negative bacteria, *P. aeruginosa* and *C. violaceum* harbor different QS systems, so QS-inhibition occurs in distinct manners. While *P. aeruginosa* harbors the four QS systems previously described, *C. violaceum* QS system consists of the CviI and CviR components. CviI synthetizes an *N*-decanoyl-*L*-homoserine lactone (C10HSL) that binds to the receptor, CviR [41]. The complex CviRC10HSL autoinduces *cviI* expression and activates the *vioA* promoter of the *vioABCDE* operon that encodes the genes for violacein production [42]. It is thought that vanillin acts as a QSI by interacting with the receptor CviR in *C. violaceum* [43]. Thus, it is possible that the extract could act as a QSI in *P. aeruginosa* while being an enhancer of the QS pathway that leads to violacein production in *C. violaceum*.

Finally, relative gene expression of several QS-related genes was quantified by RT-qPCR for *P. aeruginosa* ATCC 27853, Pa3, *S. aureus* ATCC 29213, SA007 and SA011 biofilms formed in the presence of the carboxypyrano-ant extract or carboxypyCy-3-glc, at 64 μg/mL.

Four main QS systems are present in *P. aeruginosa*, and all of them are interconnected. The LasI/LasR system regulates all other three systems, while the RhlI/RhlR and the PqsABCDE/PqsR systems regulate each other, and the AmbBCDE/IqsR system regulates the PqsABCDE/PqsR system [30]. These systems are composed by a synthase that produces the respective signal molecule and by a receptor that recognizes the signal molecule and activates the pathways that result in the expression of several genes encoding for virulence factors and biofilm formation.

In *P. aeruginosa* ATCC 27853, the carboxypyrano-ant extract significantly downregulated the transcription of *lasI* and *rhlI* and upregulated *pqsE*. The activity of both LasI/LasR and RhlI/RhlR systems results in the production of *N*-acylhomoserine lactones, which bind to the respective QS receptor, LasR or RhlR, resulting in the activation of pathways that lead to virulence factor production and biofilm formation. The release of *N*-acylhomoserine lactones seems to be inhibited upon the presence of the carboxypyrano-ant extract. In turn, the RhlI/RhlR system is known to repress/inhibit the PqsABCDE/PqsR system, thus, if the former was affected, the latter system may be activated, explaining the tendency for upregulation of *pqs* genes, in fact, with the *pqsE* gene significantly upregulated. The PqsE enzyme is a synthase of an autoinducer that activates the QS receptor RhlR [44,45]. However, herein, the *rhlR* gene was not significantly affected. Furthermore, in the Pa3 isolate, the presence of the carboxypyrano-ant extract caused upregulation of *lasR*. Because *lasR* activates the Rhl and Pqs systems [26], it would be expected that its overexpression would lead to upregulation of *rhlI*, *rhlR*, and *pqsAER*, and that was not observed, meaning that an alternative transcription regulator could be responsible for suppressing RhlR and PqsR activation, such as *algR* or *mexT*, respectively [46,47].

Regarding the effect of carboxypyCy-3-glc in the QS-related gene expression of the same strains, this compound downregulated *lasI* and *rhlI* expression and upregulated *pqsA* and *pqsE* in *P. aeruginosa* ATCC 27853, possibly by the same mechanism observed in the biofilms formed in the presence of the carboxypyrano-ant extract. Whereas in Pa3, carboxypyCy-3-glc downregulated the expression of *rhlI* and *rhlR*, indicating that this compound successfully impaired the RhlI/RhlR system in this isolate. As observed in Figure 5, *pqsAER* expression was differently affected, being somehow even opposite, in the two strains. It is possible that different strains, especially clinical isolates, may harbor mutations in some QS components, resulting in a different regulation of the QS systems. For instance, a LasR-defective mutant was detected in the lungs of a cystic fibrosis patient, and it expressed a Rhl system that acted independently of the Las system [44,48].

In *S. aureus*, the agr quorum sensing system plays a major role in the regulation of virulence factors production [10]. The agr operon consists of four genes: *agrB*, *agrD*, *agrC*, and *agrA*. Transcription of the operon is driven by the P2 promoter, which is activated by the response regulator AgrA in an autoregulated fashion. Phosphorylated AgrA also promotes transcription at the P3 promoter, leading to the expression of RNAIII. RNAIII serves to enhance the expression of genes encoding toxins, such as α-hemolysin, while reducing the expression of genes encoding surface proteins. Although most of the QS regulation depends on RNAIII, it has been shown that AgrA, upon phosphorylation, is also directly responsible for the production of some virulence factors, such as phenol-soluble modulines (PSMs) [10]. It is believed that the agr system influences biofilms in vivo by upregulating PSMs expression, which are involved in the structure of biofilms by forming channels and promoting cell dispersal from the biofilm to further colonize the host [49,50]. The staphylococcal accessory regulator (SarA) represses extracellular proteases, such as aureolysin, otherwise they would degrade PSMs while also activating the expression of RNAIII [51]. The carboxypyrano-ant extract downregulated the expression of *agrA* in the three assayed strains, even though such downregulation was only statistically significant in *S. aureus* ATCC 29213 and SA011. In the SA007 isolate, the carboxypyrano-ant extract led to upregulation of *sarA*. It would be expected that an increased *sarA* expression would result in an increased expression of RNAIII, *hla* and *ica*. The *ica* locus encodes genes required for polysaccharide intercellular adhesion (PIA) production, which is essential for biofilm formation and is under positive regulation by SarA [52]. However, it is possible that a transcription regulator like CodY could be repressing the expression of these genes, despite *sarA* upregulation [53].

Moreover, in SA011, the carboxypyrano-ant extract also decreased significantly the expression of *sarA* and *hla*, indicating that this extract could interfere as expected, by impairing the major QS components.

CarboxypyCy-3-glc upregulated *sarA* and RNAIII expression and downregulated *hla* and *ica* expression in *S. aureus* ATCC 29213. These results could possibly be explained by repression of *hla* and *ica* by the same mechanism observed in SA007 biofilms formed in the presence of the extract. Regarding the effect of carboxypyCy-3-glc in the gene expression of MDR isolates, *sarA*, RNAIII and *hla* were overexpressed in SA007, while *agrA* and *hla* were downexpressed in SA011. It has been demonstrated that AgrA negatively regulates the production of proteases that degrade several virulence factors, such as coagulases and PSMs [10,54,55]. Thus, the inhibition of AgrA in SA011 and consequent protease de-repression might be the reason why the biofilm structure is so deeply affected in the presence of either the extract or the compound. Nevertheless, we observed that the SA007 biofilm structure was also phenotypically affected by both the extract and compound; however, while the carboxypyrano-ant extract might have an impact on biofilm structure by interfering with the *ica* locus expression, which is essential for biofilm formation, the effect of carboxypyCy-3-glc in the SA007 biofilm is still to be elucidated.

Although the carboxypyrano-ant extract and carboxypyCy-3-glc could interfere with QS-related genes both in *P. aeruginosa* and *S. aureus*, there were differences among the various strains within the same species. This suggests that the effect of these compounds might be strain-specific, as specific strain lineages may have different variants of QS components [32,56,57].

Additionally, the toxic effect of the carboxypyrano-ant extract and carboxypyCy-3-glc at higher doses, such as 25 and 50 mg/kg, were evaluated in vivo using the *G. mellonella* model. CarboxypyCy-3-glc, at 50 mg/kg, was toxic to larvae; however, it revealed no negative effect on the larvae survival at 25 mg/kg. The carboxypyrano-ant extract, at both concentrations, was not toxic. In fact, since the extract is a mixture of compounds, these are present in lower concentrations, therefore, it is reasonable that the extract did not cause any toxicity at the high concentration of 50 mg/kg as did the pure compound. Thus, we could be assured that at lower doses, as those involved in the inhibition of biofilm formation and in the interference with the QS-related genes expression, the extract and the pure compound will not be toxic.

## 4. Materials and Methods

### 4.1. Synthesis of Carboxypyranocyanidin-3-O-glucoside and Carboxypyranoanthocyanins Extract

Carboxypyranocyanidin-3-*O*-glucoside (carboxypyranocy-3-glc) (Figure 10) was hemi-synthesized from the reaction of cyanidin-3-*O*-glucoside with pyruvic acid [58] and purified according to the procedure described in the literature [24]. The carboxypyranoanthocyanins extract (carboxypyrano-ant extract) was obtained from the reaction between anthocyanins present in a red wine extract with pyruvic acid according to the methodology described elsewhere [24]. The HPLC profile, at 520 nm, of the purified compound and the carboxypyrano-ant extract is present in Appendix A.

The full composition of the extract (phenolic compounds, sugar, proteins and lipids) was determined as already reported in the literature [24], and is shown in Table 2.

### 4.2. Bacterial Strains and Growth Conditions

*P. aeruginosa* ATCC 27853 and *S. aureus* ATCC 29213, as well as multidrug-resistant (MDR) clinical isolates of *P. aeruginosa* (Pa3 and PA004) and of *S. aureus* (SA007 and SA011) were used in this study. The antimicrobial resistance profile of the MDR isolates is shown in Appendix A; SA007 and SA011 are methicillin-resistant *S. aureus* (MRSA). All bacteria were grown on Mueller-Hinton (MH) agar (Liofilchem srl, Roseto degli Abruzzi, Italy) from glycerol cultures, for approximately 20 h at 37 °C, before use in the preparation of inocula in the following assays. *C. violaceum* ATCC 12472 was also used and grown in LB agar (Liofilchem srl, Roseto degli Abruzzi, Italy) and incubated at 30 °C for 48 h.

### 4.3. Minimum Inhibitory Concentration (MIC) Assay and Biofilm Formation Assay against MDR Isolates

Herein, the MIC of the carboxypyrano-ant extract and carboxypyCy-3-glc was determined against the MDR isolates (Pa3, PA004, SA007 and SA011) through the microdilution technique, using cation-adjusted Mueller–Hinton broth (MHB2, Sigma-Aldrich, St. Louis, MO, USA) as fully described by us elsewhere [24]. Their effect was also evaluated in interfering with the biofilm formation ability of those MDR isolates, following the crystal violet assay as previously described [59]. Two independent experiments were performed for each assay and each crystal violet experiment was carried out at least in triplicate (the exception was for the *P. aeruginosa* isolates, where in each only one experiment for the crystal violet assay was carried out in several replicates).

### 4.4. Microscopic Visualization of Biofilm Formation

The biofilms were grown for 24 h on µ-Dishes 35 mm high, with a polymer coverslip bottom (ibidi GmbH, Gräfelfing, Germany), from a starting inoculum of 10^6^ CFU/mL in a tryptic soy broth (TSB; Liofilchem s.r.l., Roseto degli Abruzzi, Italy), for PA004 and Pa3, or in TSBG (TSB + 1% Glucose), for SA007 and SA011. Biofilms were equally formed either in the presence of the carboxypyrano-ant extract or carboxypyCy-3-glc, each at 64 µg/mL. After 24 h at 37 °C, the planktonic phases were removed and the biofilms were rinsed with phosphate buffered saline (PBS, Sigma-Aldrich/Merck Life Science S.L.U., Algés, Portugal) and then stained with the Live/Dead staining mixture (LIVE/DEAD BacLight Bacterial Viability Kit, Thermo Fisher Scientific, Porto Salvo, Portugal) for 30 min at room temperature in the dark, rinsed again with the PBS, and then visualized under a fluorescence microscope Leica DMI6000 FFW (Leica Microsystems, Carnaxide, Portugal) or under a laser scanning confocal system Leica TCS SP5 II (Leica Microsystems, Carnaxide, Portugal), equipped with (i) an inverted microscope, Leica DMI6000-CS, using an HC PL APO CS 63× /1.30 glycerin 21 °C objective and the lasers diode 405 nm and DPSS561 561 nm; and (ii) the LAS AF software.

### 4.5. Effects on Violacein Production by C. violaceum

The effects of the carboxypyrano-ant extract and carboxypyCy-3-glc on violacein production by *C. violaceum* were assessed at 256, 64 and 16 µg/mL in a starting inoculum of 10^6^ CFU/mL in an LB broth (Liofilchem s.r.l., Roseto degli Abruzzi, Italy). Vanillin (4-hydroxy-3-methoxybenzaldehyde, Sigma-Aldrich) was included as a positive control, while an inoculum without any compound was the negative control. Following incubation at 30 °C for 24 h, the cultures were centrifuged (11,739 g, 10 min) to precipitate the insoluble violacein, and the pellet was resuspended in dimethyl sulfoxide (DMSO, Sigma-Aldrich/Merck Life Science S.L.U., Algés, Portugal). The samples were centrifuged once more (11,739 g, 10 min) and the supernatant was kept at 4 °C. Afterwards, the formation of violacein by *C. violaceum* was quantified by reverse phase liquid chromatography (Dionex Ultimate 3000, (Thermo Fisher Scientific, Porto Salvo, Portugal), in a reversed-phase C18 column (Agilent) with 250 × 4.6 mm i.d., particle size 2.7 μm and at 25 °C. The eluents used were (i) 1% (*v/v*) formic acid in water; and (ii) 0.5 % (*v/v*) formic acid in 80% (*v/v*) acetonitrile and the elution gradient was performed from 40 to 85% (ii) for 50 min at a flow rate of 0.4 mL/ min. After 50 min, the column was washed with 100% (ii) for 10 min and then it was stabilized with the initial conditions for 10 min more. The percentage of the violacein inhibition was calculated as follows (Equation (1)):(1)$$\%\;\mathrm{of}\;\mathrm{violacein}\;\mathrm{production}\;\mathrm{inhibition}\;=\frac{\mathrm{Peak}\;\mathrm{area}\;(\mathrm{LB}\;\mathrm{control})-\mathrm{Peak}\;\mathrm{area}\;(\mathrm{treated})}{\mathrm{Peak}\;\mathrm{area}\;(\mathrm{LB}\;\mathrm{control})}\text{× }100$$
and two independent experiments were performed, each allowing triplicate measurements for each condition assayed.

### 4.6. Biofilm Growth for RNA Extraction

The biofilms of the *P. aeruginosa* ATCC 27853 and Pa3, and of the *S. aureus* ATCC 29213, SA007 and SA011 were formed in TSB or TSBG, respectively, in the absence (controls) or presence of the carboxypyrano-ant extract or carboxypyCy-3-glc at 64 µg/mL. Each inoculum of approximately 10^6^ CFU/mL was dispensed into 6-well plates (2 mL/well), in 2 to 4 wells, and incubated at 37 °C for 24 h. Then, the planktonic phase of each well was removed, the biofilm was rinsed once with PBS and then collected in the PBS, washed by centrifugation (9000 g, 2 min), resuspended in RNA later (Invitrogen, Thermo Fisher Scientific, Porto Salvo, Portugal), and stored overnight at 4 °C and then at −20 °C until further RNA extraction. In the case of controls, the biofilms were collected from 1 well only, while the biofilms formed in the presence of the carboxypyrano-ant extract or carboxypyCy-3-glc were collected from 3 to 4 wells.

### 4.7. RNA Extraction

The RNA was extracted using the GeneJET RNA Purification Kit (Thermo Fisher Scientific, Porto Salvo, Portugal) according to the manufacturer’s instructions with some modifications in the first steps. Briefly, *P. aeruginosa* samples were treated with lysozyme at 4.4 mg/mL (Omega Bio-tek, Norcross, GA, USA) and incubated at room temperature for 5 min. Then, approximately 25 mg of glass beads (Omega Bio-tek, Norcross, GA, USA) were added and the tubes were vortexed at maximum speed for 5 min. The cell debris and glass beads were removed by centrifugation at 7000 g for 2 min and the supernatant was transferred to a clean RNase/DNase free microcentrifuge tube. Regarding the *S. aureus* biofilms, the samples were treated with lysozyme at 2.5 mg/mL and lysostaphin at 0.1 mg/mL (Sigma-Aldrich/Merck Life Science S.L.U., Algés, Portugal) and incubated at 37 °C for 30 min. The RNA concentration and purity were evaluated using a NanoDrop One spectrophotometer (Thermo Fisher Scientific, Porto Salvo, Portugal).

#### 4.7.1. Reverse Transcription-Quantitative Polymerase Chain Reaction (RT-qPCR) for *P. aeruginosa*

The qScript One-Step SYBR Green RT-qPCR Kit (Quantabio, Beverly, MA, USA) was used. Each 10 µL reaction contained an RNA template at 100 ng/µL, One-Step SYBR Green Master Mix, and each primer (Appendix A), at 200 nM. A ‘no template’ control (NTC) and a ‘no reverse transcriptase enzyme’ control (NRT) were included. Thermal cycling conditions were as follows: an initial reverse transcription step at 50 °C for 10 min, an activation cycle of 95 °C for 60 s, followed by 35 cycles of 95 °C for 10 s and 60 °C for 30 s. A melt curve analysis was performed at a temperature range of 69 to 95 °C with 0.3 °C/s intervals. All measurements were carried out in 3 biological replicates, each in technical duplicates.

#### 4.7.2. Reverse Transcription-Quantitative Polymerase Chain Reaction (RT-qPCR) for *S. aureus*

Regarding the *S. aureus* samples, the RT-qPCR was performed in 2 steps. First, cDNA synthesis was performed using the qScript cDNA Synthesis Kit (Quantabio, Beverly, MA, USA) and 1 µg of purified RNA. Then, in the amplification step, 10 µL PCR reaction contained 1 µL cDNA template at 5 ng/µL, 5 µL PerfeCTa SYBR Green FastMix (Quantabio, Beverly, MA, USA) each primer at 500 mM (Appendix A). An NTC and a control using the purified RNA as a template (not subjected to cDNA synthesis) were included. Thermal cycling conditions were as follows: an initial activation step of 95 °C for 10 min, followed by 40 cycles of 95 °C for 15 s and 60 °C for 60 s. A melt curve analysis was performed at a temperature range of 69 °C to 95 °C with 0.3 °C/s intervals. All measurements were carried out in 3 biological replicates, each in technical duplicates.

### 4.8. Galleria Mellonella Larvae Toxicity Assay

The greater wax moth larvae were obtained from Biosystems (Biosystems Technology, Crediton, UK). Groups of 16 larvae in the final instar stage weighing between 180 and 350 mg were used for each experimental condition. The larvae were injected in the last left proleg with 10 µL of the carboxypyrano-ant extract or carboxypyCy-3-glc at 25 and 50 mg/kg. Controls included a group of larvae that did not receive any injection and another group of larvae that were injected with 10 µL of PBS. The larvae were then incubated in the dark at 37 °C and they were assessed daily for survival for up to 7 days post-injection. The larvae were considered dead when no movement was displayed in response to any stimulation by touch. Two independent experiments were performed for each condition.

### 4.9. Statistical Analysis

The results regarding the inhibition of biofilm formation, *C. violaceum* assay and toxicity assessment in *G. mellonella* are expressed as mean values ± standard error of the mean (SEM). The statistical significance of differences between controls and experimental groups was determined using Student’s *t*-test and *p* values < 0.05 were considered statistically significant.

Regarding the RT-qPCR analyses, the Q software (Quantabio, Beverly, MA, USA) generated the cycle-threshold (Ct) values, and then the relative levels of gene expression were calculated by the ∆∆Ct method, using the *rpoS* or *16S* rRNA genes as internal controls (reference genes) for *P. aeruginosa* or *S. aureus*, respectively, as follows (Equations (2)–(4)):∆Ct (control) = Ct Gene of interest (control) − Ct Reference Gene (control) (2)
∆Ct (treated) = Ct Gene of interest (treated) – Ct Reference Gene (treated), (3)
∆∆Ct = ∆Ct (treated) − ∆Ct (control),(4)
results are expressed as ∆∆Ct ± confidence interval at 95% confidence. If the confidence interval does not include the null value (zero), then we concluded that there was a statistically significant difference between the groups. The statistical analyses were performed in Microsoft Excel and RStudio.

## 5. Conclusions

In this study, we have explored one pyranoanthocyanin, carboxypyCy-3-glc, and one extract, carboxypyrano-ant extract, and found that they both could (i) strongly impair the biofilm formation by *S. aureus* multidrug-resistant isolates, while not evidently affecting *P. aeruginosa* biofilm formation; (ii) interfere with QS signaling in both *P. aeruginosa* and *S. aureus*, even if at different levels; and (iii) show low toxicity in the in vivo larvae model of *G. mellonella*. Although their mechanism of action is still to be unraveled, these compounds appear to be promising in interfering with the QS system of both susceptible and MDR strains of *P. aeruginosa* and *S. aureus*. Nonetheless, more in-depth studies (e.g., proteomics, transcriptomics, in silico analysis) need to be conducted with several other pathogenic bacterial strains, susceptible and MDR, to assess the potential of these compounds in inhibiting QS, and by which mechanism, and thus in impairing virulence factors production and biofilm formation.

The obtainment process of an extract is simpler and shorter than that of a pure compound, and therefore, could be an advantage for the extract use over the pure compound when both have a similar activity. The carboxypyrano-ant extract interfered with the QS systems of both *P. aeruginosa* and *S. aureus*, thus, it may hold potential in further research and development of novel products aiming at effectively restraining the pathogenicity of strains responsible for causing biofilm-associated cutaneous infections, particularly those related to chronic infected wounds.

## 6. Patents

Provisional patent application N. 117058: Process for extraction and hemi-synthesis of pyranoanthocyanins and skincare cosmetic formulations containing them.

## Figures and Tables

**Figure 1 ijms-22-08559-f001:**
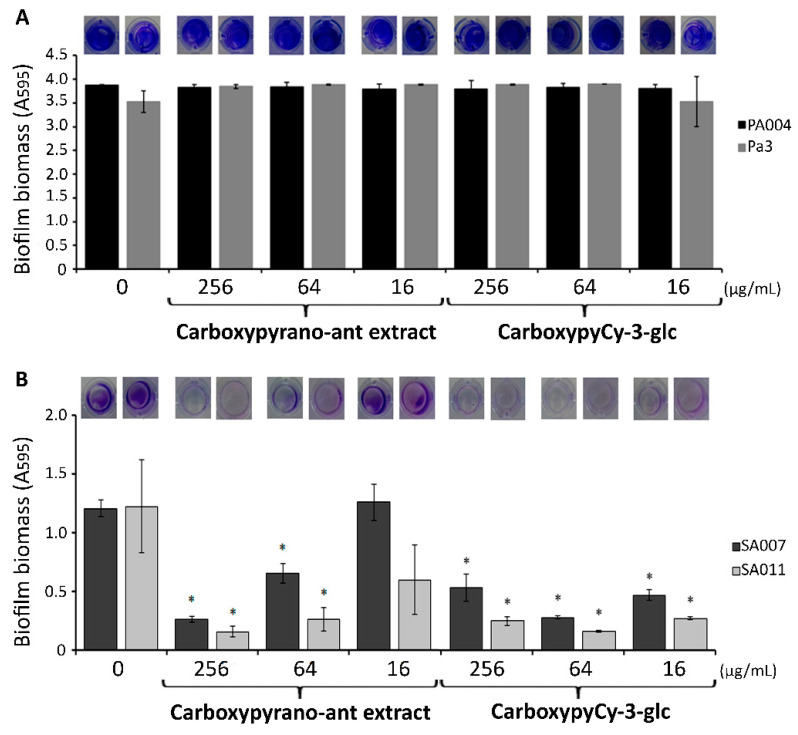
Biomass quantification using the crystal violet assay of (**A**) PA004 and Pa3 and (**B**) SA007 and SA011 biofilms formed in presence of carboxypyrano-ant extract and carboxypyCy-3-glc at 256, 64 and 16 μg/mL, and in absence of any compound or extract (control). Error bars represent SEM. Differences between the experimental groups and the respective controls were statistically significant for * *p* < 0.05.

**Figure 2 ijms-22-08559-f002:**
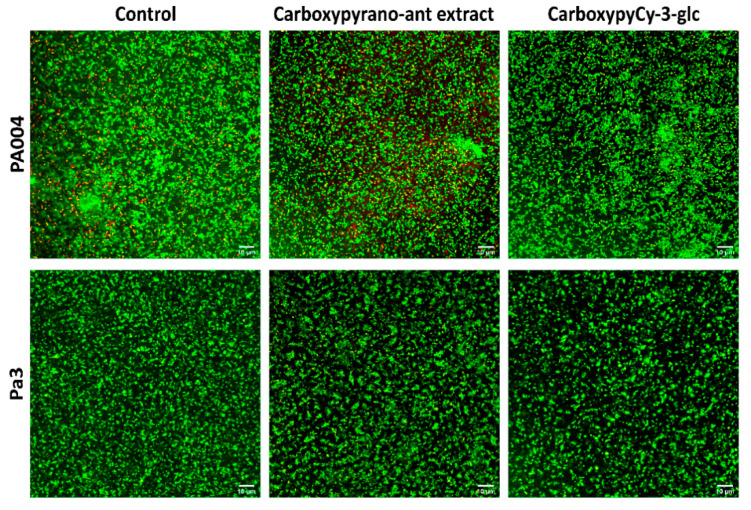
Confocal laser scanning microscopy images, after Live/Dead staining, of PA004 and Pa3 biofilms formed for 24 h in absence (control) and presence of carboxypyrano-ant extract or carboxypyCy-3-glc, at 64 µg/mL. Scale bar: 10 µm.

**Figure 3 ijms-22-08559-f003:**
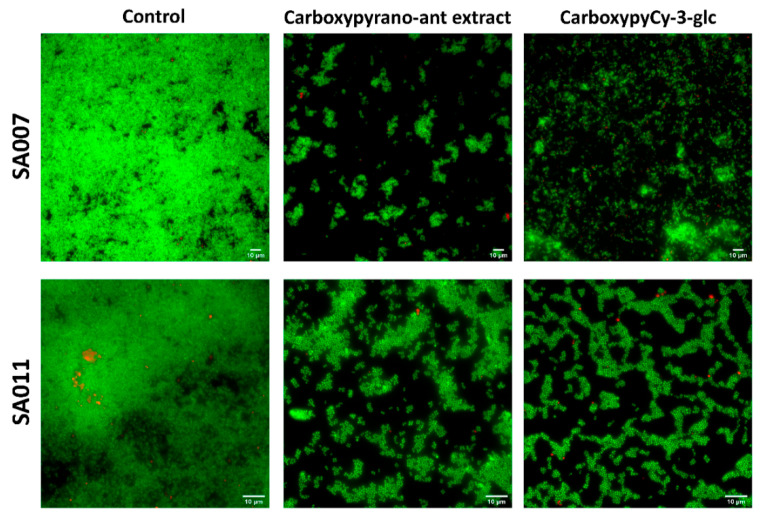
Fluorescence microscopy representative images, after Live/Dead staining, of SA007 and SA011 biofilms formed for 24 h in absence (control) and presence of carboxypyrano-ant extract or carboxypyCy-3-glc, at 64 µg/mL. Scale bar: 10 µm.

**Figure 4 ijms-22-08559-f004:**
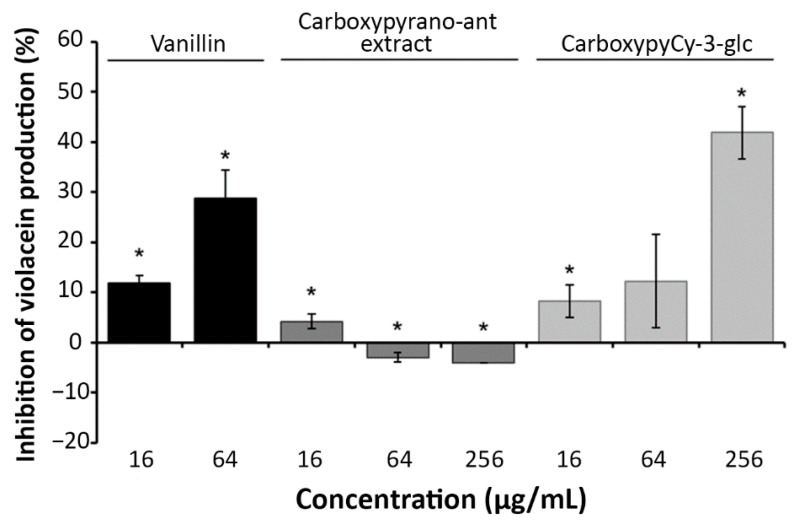
Effect of carboxypyrano-ant extract and carboxypyCy-3-glc on violacein production by *C. violaceum* ATCC 12472. Vanillin, a known QSI, was included as a positive control. Error bars represent SEM. Differences between the experimental groups and the respective controls were statistically significant for * *p* < 0.05.

**Figure 5 ijms-22-08559-f005:**
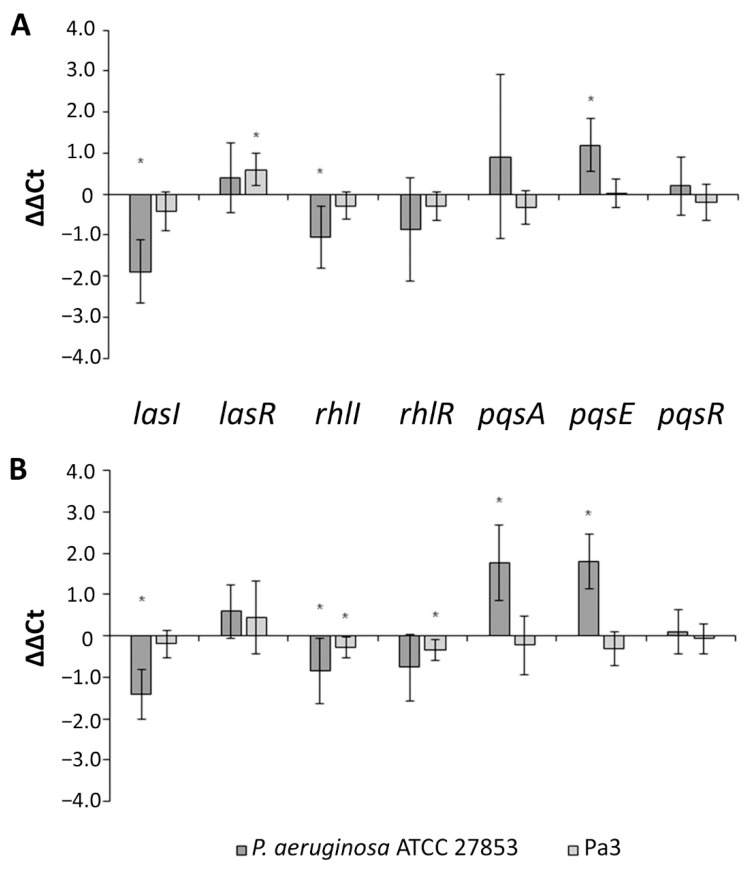
Relative quantification of the QS-related genes *lasI*, *lasR*, *rhlI*, *rhlR*, *pqsA*, *pqsE*, *pqsR* in *P. aeruginosa* ATCC 27853 and Pa3 biofilms formed in the presence of carboxypyrano-ant extract (**A**) or carboxypyCy-3-glc (**B**), at 64 μg/mL. Results are expressed as ΔΔCt ± confidence interval at 95% confidence. Differences between the experimental group and the control (absence of extract or compound) were statistically significant (*) when the confidence interval does not contain the zero.

**Figure 6 ijms-22-08559-f006:**
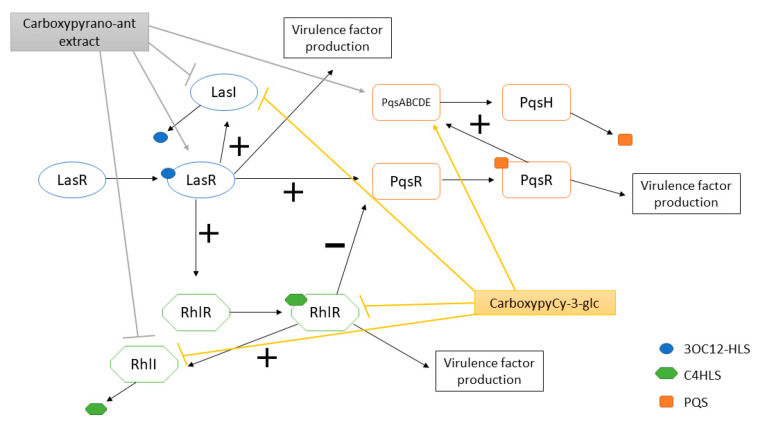
LasI/LasR, RhlI/RhlR and Pqs systems of *P. aeruginosa* and overview of the effects of carboxypyrano-ant extract and carboxypyCy-3-glc on the QS systems of *P. aeruginosa* ATCC 27853 and/or Pa3. (+): activation; (−): inhibition.

**Figure 7 ijms-22-08559-f007:**
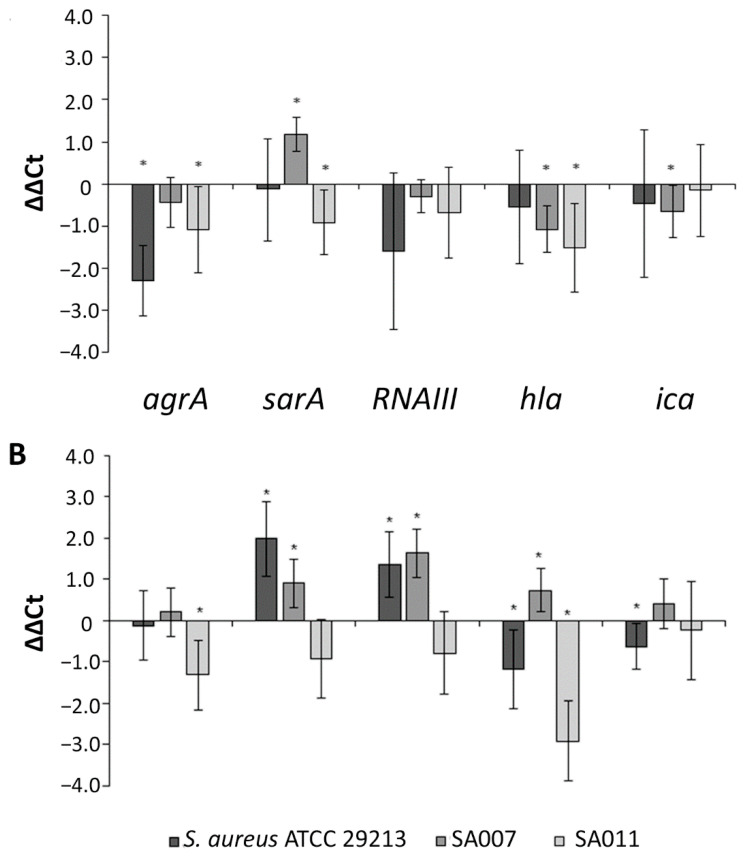
Relative quantification of the QS-related genes *agrA*, *sarA*, RNAIII, *ica*, *hla* in *S. aureus* ATCC 29213, SA007 and SA011 biofilms formed in the presence of carboxypyrano-ant extract (**A**) or carboxypyCy-3-glc (**B**), at 64 μg/mL. Results are expressed as ΔΔCt ± confidence interval at 95% confidence. Differences between the experimental group and the control (absence of extract or compound) were statistically significant (*) when the confidence interval does not contain the zero.

**Figure 8 ijms-22-08559-f008:**
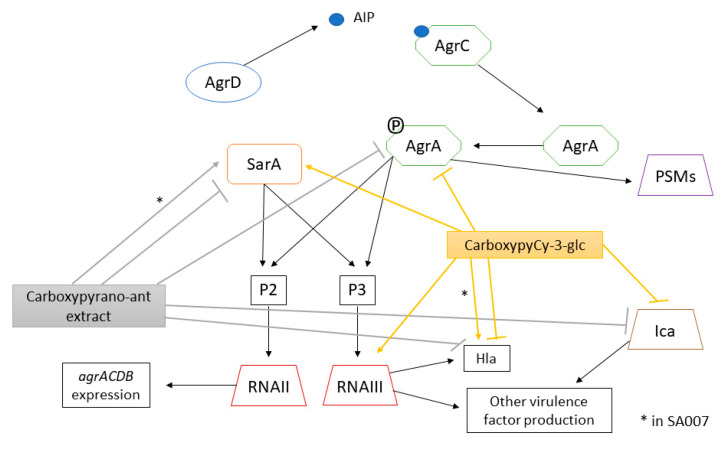
Agr system of *S. aureus* and overview of the effects of carboxypyrano-ant extract and carboxypyCy-3-glc on the QS system of *S. aureus* ATCC 29213 and/or SA007 and SA011. (*) points out the opposite effects of the extract or compound on the same gene among the strains.

**Figure 9 ijms-22-08559-f009:**
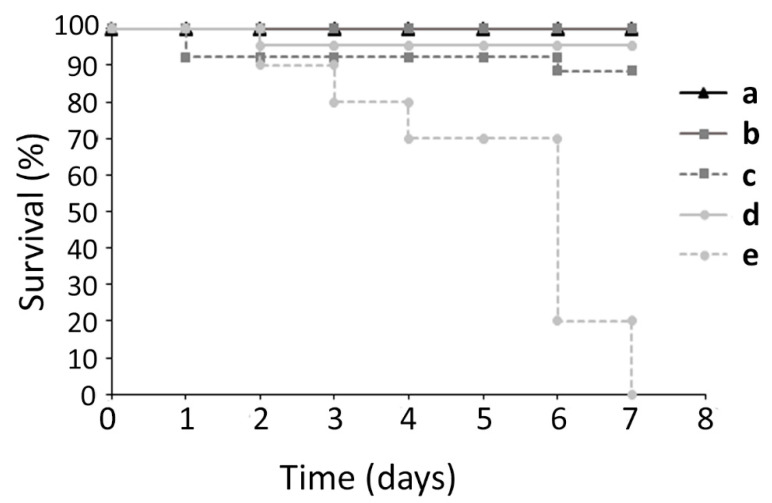
Kaplan-Meier survival-curve of *G. mellonella* larvae injected with: (a) PBS, (b) carboxypyrano-ant extract at 25 mg/kg, (c) carboxypyrano-ant extract at 50 mg/kg, (d) carboxypyCy-3-glc at 25 mg/kg, and (e) carboxypyCy-3-glc at 50 mg/kg.

**Figure 10 ijms-22-08559-f010:**
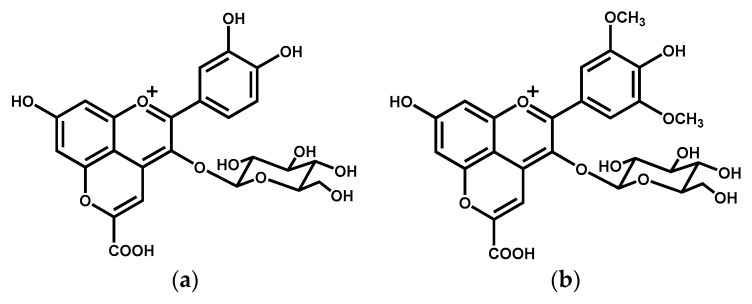
Chemical structures of (**a**) carboxypyranocyanidin-3-*O*-glucoside (carboxypyCy-3-glc) and (**b**) carboxypyranomalvidin-3-*O*-glucoside (carboxypyMv-3-glc), the main pyranoanthocyanin present in the carboxypyrano-ant extract.

**Table 1 ijms-22-08559-t001:** Minimum inhibitory concentration (MIC) and minimum bactericidal concentration (MBC) values (μg/mL) of carboxypyrano-ant extract and carboxypyCy-3-glc against PA004, Pa3, SA007 and SA011.

	PA004	Pa3	SA007	SA011
	MIC (MBC)			
Carboxypyrano-ant extract	>512 (−)	512 (>512)	>512 (−)	>512 (−)
CarboxypyCy-3-glc	>512 (−)	>512 (−)	>512 (−)	>512 (−)

(−) not determined.

**Table 2 ijms-22-08559-t002:** Full composition of the carboxypyrano-ant extract.

Components	(mg/mg ± SD)
Pyranoanthocyanins	0.18 ± 0.002
Low weight polyphenols	0.010 ± 0.002
Flavonols	0.056 ± 0.006
Protein content	0.096 ± 0.002
Total lipids	0.05 ± 0.01
Sugar content	0.0184 ± 0.0023

## Data Availability

The data presented in this study are openly available in [FigShare] at [doi], reference number [10.6084/m9.figshare.14912460].

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
