# Peer review of "Pyranoanthocyanins Interfering with the Quorum Sensing of Pseudomonas aeruginosa and Staphylococcus aureus"

_ijms, 2021, doi:10.3390/ijms22168559_

Round 1

Reviewer 1 Report

Bessa et al. investigated the ability of carboxypyranoant extract of red wine, and carboxypyranocyanidin-3-O-glucoside to inhibit biofilm formation and the quorum sensing of both P. aeruginosa and S. aureus. The authors showed that the extract and pure compound could significantly inhibit the biofilm formation of S. aureus but had no considerable effect on the biofilm formation of P. aeruginosa. Interestingly, the authors showed that carboxypyranocyanidin-3-O-glucoside interfere with the expression of the quorum sensing-related genes in both  P. aeruginosa and S. aureus. Finally, the authors have investigated the toxicity of the extract and carboxypyranocyanidin-3-O-glucoside which revealed that both extract and compound have no significant toxicity up to 50mg/kg. Although arboxypyranocyanidin-3-O-glucoside is well known/studied in literature as bioactive/antimicrobial agent, this study show for the first time that the antimicrobial activity of this class of compounds could be attributed to their ability to inhibit the QS system. The authors proved their hypothesis by performing qPCR.

Although, this is quite interesting findings,/study, the authors did not show any possible mechanism of this effect. Most of the performed experiemnts are very simple and does not give a full conclusion of the study. The authors need to address many points before this study would be suitable for publication. The following are detailed concerns;

- the introduction is missing many recent and relevant studied about this compd. e.g,

The Role of Anthocyanins, Deoxyanthocyanins and Pyranoanthocyanins on the Modulation of Tyrosinase Activity: An In Vitro and In Silico Approach; https://doi.org/10.3390/ijms22126192

Cyanidin-3-glucoside Lipophilic Conjugates for Topical Application: Tuning the Antimicrobial Activities with Fatty Acid Chain Length; https://doi.org/10.3390/pr9020340

- the authors should do some extra experiemts to reveal/indicate the mechanism of QS inhibition. e.g, agr and fsr systems? this is very easy to test and would be informative.

- the above concern, could be also supported by in silico study. Since, several protocols have been reported for in silico docking for this compound, it would be straight forward to evaluate the binding of this compd toward several targeted proteins in QS system.

- Although the authors have followed reported study for extraction/hemi-synthesis, the authors have to provide HPLC analysis/spectra of their materials that they have used through their study.

5- The toxicity experiment is poorly designed and not fully informative. What is LD for the pure compound? it is obvious that the compd is highly toxic at 50mg/kg which indicate a really low LD value!!!

6- One drawback of most experiments is missing of control?? e.g the effect on QS-related genes expression??

7- why the authors expressed the inhibition in ug/mL not in M?

Reviewer 2 Report

Overall comment: 

The manuscript was overall well-written but had an overall unusual layout where results and discussion were mixed. It is recommended to move the discussion to a Discussion section before the conclusions.  The manuscript needs to be revised to fit the more traditional format

General comment:

First time a bacterial name is used, use the full genus species (eg Pseudomonas aeruginosa) and then on subsequent uses abbreviate (P. aeruginosa). Please adjust bacterial names to be consistent

Specific comments: 

Lines 132-133: describe and reference the methods used for MIC testing

Line 169: 100% of what?

Table 2: what does (-) mean for MBC values? Was it not tested or not determined? Please add info

Figure 2: state what error bars represent in the figure legend.

Lines 164-271: these lines would be better in the discussion rather than the results

Line 286: define CLSM 

Lines 307-317: should be moved to the discussion and not stay in results

Figure 5: State what error bars represent

Lines 335-346: should be moved to discussion and not be part of the results section

Lines 376-402: should be in a discussion

Figure 8: the shades of grey that are chosen for the different lines are very hard to distinguish. If authors wish to keep lines grey, at leas use a different symbol for each condition so that they can be better visualized.

Schemes 1 and 2 should be figures, not schemes

Round 2

Reviewer 1 Report

thanks to the authors for addressing most of the concerns highlighted by the reviewer. The manuscript has been significantly improved.

  • The only missing point, the HPCL analysis was not attached to the SI.

Author Response

Dear reviewer,

Thank you for your positive feedback. 

I apologize, because I think you were not able to view the revised version of the Supplementary Material file, which I have uploaded together with the revised version of the manuscript. We have inserted there a Figure with the HPLC results. We upload again a new version of the Supplementary Material file.